# Graph-Constrained Structure Search for Tensor Network Representation

## Abstract

Recent works paid effort on the structure search issue for tensor network (TN) representation, of which the aim is to select the optimal network for TN contraction to fit a tensor. In practice, however, it is more inclined to solve its sub-problem: searching TN structures from candidates with a similar topology like a cycle or lattice. We name this problem *the graph-constrained structure search*, and it remains open to this date. In this work, we conduct a thorough investigation of this issue from both the theoretical and practical aspects. Theoretically, we prove that the TN structures are generally irregular under graph constraints yet can be universally embedded into a low-dimensional regular discrete space. Guided by the theoretical results, we propose a simple algorithm, which can encode the graph-constrained TN structures into fixed-length strings for practical purposes by a "random-key" trick, and empirical results demonstrate the effectiveness and efficiency of the proposed coding method on extensive benchmark TN representation tasks.

## 1 Introduction

Tensor networks (TNs) are recognized as a popular framework for solving extremely high-dimensional problems arising in domains such as quantum simulation, machine learning and signal processing. In general, TNs are used to represent the high-dimensional states/models/data by a network of low-dimensional tensors (*a.k.a.*, cores), such that the requirement on computation and storage would be significantly reduced.

It is of importance to select an appropriate structure in the practical use of TNs. There are many studies on learning TN ranks for specific models [26–28, 43, 46, 47] to name a few, and recently several works paid the effort on learning more general TN structures with arbitrary topology [17, 19, 21, 25]. Surprisingly, however, none of them can effectively solve a seemly easier task: *how to learn the optimal matching from the modes onto the cores of a TN?* For instance as illustrated in Figure 1, there are three different candidates to represent a tensor by tensor ring (TR) [47]. We need algorithms, which can learn the optimal one from the three. It is actually a special case of learning the optimal TN structures under *graph constraints*, a sub-problem of the existing structure search for TN representation.

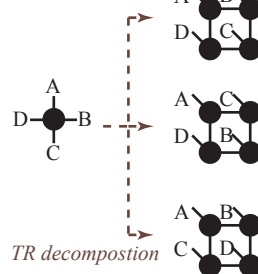

Figure 1: Which tensor ring (TR) is the optimal?

The state of affairs raises important unresolved questions. *Is the aforementioned task really easier than the general structure search? What are the properties of TN structures under graph constraints, and how to effectively solve the problem in practice?*

In this work, we shed light on these questions through a theoretical and empirical investigation of the graph-constrained TN structures.

We first prove the graph constraint makes TN structures being irregular. In particular, both the addition and random perturbation is not closed on the candidate set. This result helps to explain why the conventional search algorithms on grids give no guarantee of feasibility of the solutions. Furthermore, on the scale of the search problem, we prove the *symmetry* of the graph-constraint plays a role to determine the cardinality of TN structures, yet there exists a universal cardinality bound across a varies of practical TNs, such as tensor train (TT) [30], tensor ring (TR) and PEPS [38]. The result reveals the possibility to construct a regular discrete space, from which we can represent those irregular TN structures by elements in a compact manner.

Guided by the theoretical results, this work also sheds light on a practical solution for the graph-constrained structure search issue. We propose a novel coding method to encode TN structures into fix-length strings by a "random-key" trick, a random mapping from TN structure space to coding space. The regularity of the coding space allows to apply the population-based algorithms equipped with the proposed coding method to tackling the search issue for TN representation effectively. We conduct extensive experimentation on a variety of benchmarks. The results show that the proposed method often obtain better TN structures than many existing rank-selection and structure search algorithms.

## 2 Preliminaries and problem setup

In this section, we present the basic concepts on tensor network (TN), and give a formal definition of the *graph-constrained* structure search for TN representation.

### 2.1 Tensor network (TN) and structure search for tensor network representation (TNR)

An order-$N$ *tensor* is a multi-dimensional array of real numbers represented by $\mathcal{X}_{i_1,i_2,\dots,i_N} \in \mathbb{R}^{I_1 \times I_2 \times \cdots \times I_N}$, where $i_m$, $m \in [N]$ is defined as the *index* regarding the $m$th mode of $\mathcal{X}^1$ and $[N]$ denotes a set of integers from 1 to $N$. *Tensor contraction* [10], a binary operation on tensors, is defined as a multiplication of two tensors under their same indices. For instance, given two order-2 tensors $\mathcal{A}_{i,j} \in \mathbb{R}^{I \times J}$, $\mathcal{B}_{j,k} \in \mathbb{R}^{J \times K}$, the tensor contraction of $\mathcal{A}$ and $\mathcal{B}$ under the index $j$ returns $\mathcal{C}_{i,k} = \mathcal{A}_{i,j}\mathcal{B}_{j,k} \in \mathbb{R}^{I \times K}$, which is equivalent to the matrix multiplication.

A *tensor network (TN)* is roughly defined as a collection of tensors (*a.k.a.*, cores), which are tensor-contracted under some, or all, of their indices according to a specific pattern [29]. Recent works [25, 42] show that the "patterns" of TNs can be precisely described by edge-weighted simple graphs. TN structures thus can be formulated by adjacency matrices of graphs. Formally, we define the TN with a general "pattern" as follows.

**Definition 1 (Tensor network.)** *Let* $\mathbb{A}_R = \left\{ \mathbf{A} \in (\mathbb{Z}_{R+1})^{N \times N} \,|\, \mathbf{A}(i,i) = 0, \, \forall i \in [N], \, and \, \mathbf{A} = \mathbf{A}^\top \right\}$, *an order-$N$ tensor network (TN) of the size $I_1 \times I_2 \times \cdots \times I_N$ under a structure $\mathbf{A} \in \mathbb{A}_R$ defines a mapping :*

$$\mathcal{X} = TN(\mathbb{V}; \mathbf{A}) \in \mathbb{R}^{I_1 \times I_2 \times \cdots \times I_N}, \tag{1}$$

*where* $\mathbb{V} = \{\mathcal{V}_i, \, i \in [N]\}$ *represents a collection of cores in which the size of $\mathcal{V}_i$, $i \in [N]$ equals the multiplication of $I_i$ and all non-zero entries of $\mathbf{A}(i,:)$, and $TN(\,\cdot\,; \mathbf{A})$ denotes a series of tensor contractions of $\mathbb{V}$ under the indices [25] described by $\mathbf{A}$.*

We observe that Definition. 1 models a rich family of TNs with the ranks upper-bounded by $R$ (due to $\mathbb{Z}_{R+1}$), including TT, TR, PEPS and *etc.*, but also note that the TNs that contain internal cores are not included in this form.

Tensor network representation (TNR) of a tensor $\mathcal{X}$ is defined as finding a specific core-set $\mathbb{V}$ such that Eq. (1) holds. The *structure search for TNR* is thus to find the optimal matrix $\mathbf{A} \in \mathbb{A}_R$, such that $\mathcal{X}$ can be represented by $\mathbb{V}$ that satisfies Eq. (1). In particular, the search problem can be solved by

$$\min_{\mathbf{A} \in \mathbb{A}_R} \phi_{\mathcal{X}}(\mathbf{A}), \quad s.t. \, \mathcal{X} = TN(\mathbb{V}; \mathbf{A}) \, for \, some \, \mathbb{V}, \tag{2}$$

---

[1]The indices would be omitted for brevity if there is no confusion.

where $\phi_{\mathcal{X}} : \mathbb{A}_R \to \mathbb{R}$ denotes a measure of the TN structures like compression ratio. Note that similar frameworks were also introduced in works [17, 21], where the entries of $\mathbf{A}$ corresponds to the TN-ranks formulated as a vector in those works. Lemma 5 in Sec. 3 will show the matrix form of $\mathbf{A}$ would provide additional structural information to analyse the property of TN structures.

## 2.2 Graph-constrained structure search for TNR

The graph-constrained structure search issue is also modeled as (2) yet constraining the feasible space $\mathbb{A}_R$ into a graph-induced subset, in which the TN structures have similar topological forms. To build the connection to graphs, we first show the existence of a bijective mapping from $\mathbb{A}_R$ to a graph space.

**Lemma 2** *There is a bijective mapping* $\Psi : \mathbb{A}_R \to \mathbb{G}_R$, *where* $\mathbb{G}_R$ *denotes a set containing all possible vertex-labeled, simple yet weighted graphs* $(G, f_R) = (V, E, f_R)$ *with* $N$ *vertices and a edge-weighting function* $f_R : e \in E \to [R]$, *and we name the unweighted part* $G$ *the* topology *of TN a TN structure.*

The claim is naturally true by the relation between graphs and the adjacency matrices. The bijection in Lemma 2 implies that for each $\mathbf{A} \in \mathbb{A}_R$ we can always find a unique $(G, f_R)$ corresponding to it. Table 1 in the *supplementary material* illustrates the correspondence between graphs and the well-known TNs. We next construct graph-constrained TN structures by the isomorphism of a graph $G_0 = (V, E_0)$ and the mapping $\Psi$ given in Lemma 2. A formal definition is given as follow.

**Definition 3 (Graph-constrained TN structures.)** *Given a vertex-labeled simple graph* $G_0 = (V, E_0)$ *and the mapping* $\Psi$ *in Lemma 2, the TN structures under* $G_0$ *are defined as*

$$\mathbb{H}_{G_0,R} = \{\mathbf{H} \in \mathbb{A}_R | G_H \cong G_0 \text{ where } (G_H, f_{R,H}) = \Psi(\mathbf{H})\}, \tag{3}$$

*where* $\cong$ *denotes the graph isomorphism.*

As given in Definition 3, $\mathbb{H}_{G_0,R}$ is a subset of $\mathbb{A}_R$ and its elements own the topologies being isomorphic to $G_0$. For instance, suppose $G_0$ to be a cycle graph of 4 vertices, *i.e.*, $C_4$, then $\mathbb{H}_{C_4,R}$ contains all TR structures of order-4 with the ranks upper-bounded by $R$ as Figure 1. It is thus expected to solve the mentioned optimal matching problem by searching structures on $\mathbb{H}_{G_0,R}$. Not only that, but also note $\mathbb{H}_{G_0,R}$ equals $\mathbb{A}_R$ if $G_0$ is a completion graph, *i.e.*, $K_N$. Next, we define the problem of graph-constrained structure search for TNR by $\mathbb{H}_{G_0,R}$.

**Definition 4 (Graph-constrained structure search for TNR.)** *Given a graph* $G_0$ *and the corresponding* $\mathbb{H}_{G_0,R}$ *obtained as Definition 3, the graph-constrained structure search for TNR is to solve the following problem:*

$$\min_{\mathbf{H} \in \mathbb{H}_{G_0,R}} \phi_{\mathcal{X}}(\mathbf{H}), \quad s.t. \, \mathcal{X} = TN(\mathbb{V}; \mathbf{H}) \, for \, some \, \mathbb{V}. \tag{4}$$

It is shown from Definition 4 that the set $\mathbb{H}_{G_0,R}$ restricts the optimization process only searching on the TN structures, which has the same topology $G_0$ up to *permutations of the vertices* [5]. Moreover, although (4) owns a similar form to its unconstrained counterpart (2), it will be proved in the next section that the existing algorithms on (2) may be not available on the graph-constrained search issue.

**Remark.** Note that Definition 1 allows the entries of $\mathbf{A}$ to equal 1, which implies the rank-one contraction between cores. According to the fact given in [25, 42] that the weight-one edges can be removed from TNs, we thus know that the solution of (4) would have a *subgraph* of $G_0$ as its "true" topology. Therefore, solving (4) has the capability of achieving not only isomorphic but also subgraphs of $G_0$.

## 3 Algebraic properties of graph-constrained TN structures

In this section, we focus on the properties of $\mathbb{H}_{G_0,R}$, the set containing all TN structures constrained under a graph $G_0$. From an algebraic perspective, we first show $\mathbb{H}_{G_0,R}$ is irregular under most of graph constraints by proving that the set is not closed under addition and random perturbation. After that, we analyse the cardinality of $\mathbb{H}_{G_0,R}$, which reflects the scale of the search problem. We derive

126  the precise cardinality of $\mathbb{H}_{G_0,R}$ across many well-known TNs, and prove a universal cardinality
127  bound of $\mathbb{H}_{G_0,R}$ under all connected low-degree graphs.

128  To understand the property of $\mathbb{H}_{G_0,R}$, we first prove all its elements own a factorization of the
129  multiplication of a rank-induced matrix and a permutation matrix.

130  **Lemma 5 (Factorization of $\mathbb{H}_{G_0,R}$.)** *Given a vertex-labeled simple graph $G_0 = (V, E_0)$, for any*
131  $\mathbf{H} \in \mathbb{H}_{G_0,R}$, *there exists a permutation matrix $\mathbf{P}$ of the size $|V| \times |V|$ and a bijective linear mapping*
132  $\Omega_{G_0} : (\mathbb{Z}_{R+1})^{|E_0|} \to (\mathbb{Z}_{R+1})^{|V| \times |V|}$ *such that $\mathbf{H}$ can be factorized as*

$$\mathbf{H} = \mathbf{P}\Omega_{G_0}(\mathbf{r})\mathbf{P}^\top, \tag{5}$$

133  *where $|\cdot|$ denotes the cardinality and $\mathbf{r} \in (\mathbb{Z}_{R+1})^{|E_0|}$ denotes the rank vector of dimension $|E_0|$.*

134  Intuitively, Lemma 5 implies that the rank-induced matrix $\Omega_{G_0}(\mathbf{r})$ forms a linear sub-space of
135  dimension $|E_0|$, then $\mathbb{H}_{G_0,R}$ takes all "flips and rotations" of the subspace into account due to the
136  permutation matrix $\mathbf{P}$. A visual illustration of $\mathbb{H}_{G_0,R}$ is shown on the most left of Figure 2. We can
137  see that $\mathbb{H}_{G_0,R}$ has an "irregular shape" visually, and this property is formally proved as follows.

138  **Proposition 6 (Irregularity of $\mathbb{H}_{G_0,R}$.)** *Assuming $R \geq 2$, the following two claims are held.*

139      *1. Addition (modulo $R + 1$) is not closed on $\mathbb{H}_{G_0,R}$ if $G_0 = (V, E_0)$ or its complement is not*
140        *complete;*
141      *2. With a relatively sparse graph $G_0$, the Bernoulli-distributed perturbation on $\mathbb{H}_{G_0,R}$ is not*
142        *closed with a probability approximately being larger than $(1 - 1/R)^{|E_0|}$.*

143  The proof is given as *supplementary material*. Proposition 6 effectively say that the operations used
144  in common search algorithms, such as the recombination and mutation in genetic algorithms (GAs)
145  or progressive search in greedy methods, cannot guarantee the outputs being contained by $\mathbb{H}_{G_0,R}$,
146  leading to the invalidation of those algorithms on this issue.

147  Next, we jump to the cardinality of $\mathbb{H}_{G_0,R}$, which reflects how many candidates we have under a
148  graph constraint. From a information-theoretic perspective, the cardinality is proportional to the least
149  required code length on TN structures in general. A smaller cardinality generally implies a easier
150  search process especially for the population-based algorithms. Below, we first prove the cardinality
151  of $\mathbb{H}_{G_0,R}$ under a general graph constraint.

152  **Lemma 7 (Cardinality of $\mathbb{H}_{G_0,R}$.)** *Given a vertex-labelled simple graph $G_0 = (V, E_0)$, we have*

$$\log(|\mathbb{H}_{G_0,R}|) = |E_0| \log(R) + \log(|V|!) - \log(|Aut(G_0)|), \tag{6}$$

153  *where $\log(\cdot)$ denotes the natural logarithm and $Aut(G_0)$ denotes the graph automorphisms of $G_0$.*

154  As shown on the right of Eq. (6), the first two terms correspond to the TN-ranks and permutations as
155  Lemma 5, respectively, while the third term $\log(|Aut(G_0)|)$ reflects the *symmetry* of $G_0$. it implies
156  the cardinality of $\mathbb{H}_{G_0,R}$ would be small if $G_0$ owns strong symmetry. From the TN perspective, it
157  means the TNs with symmetric topologies like TR and the complete TN (CTN) [48] are expected to
158  own a smaller size of $\mathbb{H}_{G_0,R}$. For those well-known TNs, we show their corresponding cardinality of
159  $\mathbb{H}_{G_0,R}$ as follow.

160  **Proposition 8** *Assume order-$N$ TN models, of which the ranks are upper-bounded by $R$, then we*
161  *have*

    162  *1. **TT** [30]: $\log(|\mathbb{H}_{P_N,R}|) = (N - 1) \log(R) + \log(N!) - \log(2)$*
    163  *2. **TR** [47]: $\log(|\mathbb{H}_{C_N,R}|) = N \log(R) + \log((N - 1)!) - \log(2)$*
    164  *3. **CTN** [48]: $\log(|\mathbb{H}_{K_N,R}|) = (N^2 - N) \log(R)/2$*
    165  *4. **T-tree** [42]: $(N - 1) \log(R) + \log(N) \leq \log(|\mathbb{H}_{T_N,R}|) \leq \log(|\mathbb{H}_{P_N,R}|)$*
    166  *5. **PEPS** [38] : $\log(|\mathbb{H}_{L_{m,n}}|) \leq (2mn - m - n) \log(R) + \log((mn)!) - \log(4)$*
    167  *6. **Tucker**[2] [36]: $\log(|\mathbb{H}_{K_{1,N}}|) = N \log(R)$*

---

[2]Note that the Tucker model is not strictly contained by Definition. 1.

In Proposition 8, the inequalities for the T-tree models is due to the variety of the tree structures, and in PEPS the equality is held if $m$ and $n$ are relatively prime. We observe from Proposition 8 that TR would have a smaller $\mathbb{H}_{G_0,R}$ than TT in the case of large $N$. It is intuitively true since the TR structure is more symmetric than the one of TT. However, we also observe that, except CTN and the Tucker model, there always exists a factorial of $N$ in the equations for the rest of TNs. It implies that the cardinality of $\mathbb{H}_{G_0,R}$ for those TNs is *not significantly different from each other.* Below, we prove the fact is true for all TNs, of which the corresponding $G_0$ is connected and low-degree.

**Proposition 9 (A universal cardinality bound on $\mathbb{H}_{G_0,R}$.)** *Assume $G_0 = (V, E_0)$ is connected graph and its maximum degree $\Delta_{G_0}$ is a constant that is far less than $|V|$, then we have*

$$\log(|\mathbb{H}_{G_0,R}|) \geq \mathcal{O}\left(|V|\log(R) + |V|\log(|V|)\right), \tag{7}$$

*where $\mathcal{O}(\,\cdot\,)$ denotes the big-O notation.*

The result is proved by bounding the both $|E_0|$ and $|Aut(G_0)|$ in Lemma 5 by the maximum degree $\Delta_{G_0}$ using the Handshaking lemma known in graph theory and Theorem 2 given in [22], respectively. In addition, we also use the Stirling's approximation [32] to obtain a tight bound for the logarithm of factorials to further simplify the expression.

The assumption of a small $\Delta_{G_0}$ is reasonable since in the practical TNs the cores are expected to be low-order (see Table 1 given in the supplementary material for instance). Proposition 9 means that there is a $G_0$-*independent bound* on the cardinality of $\mathbb{H}_{G_0,R}$ for all connected and low-degree graphs, and we can see the bound is relatively tight by intuitively comparing the results with Proposition 8.

As shown in (7), the first term $|V|\log(R)$ corresponds to the number of all possible ranks bounded by $R$, and the second term $|V|\log(|V|)$ has the same scale to $\log(|V|!)$ for the Stirling's approximation. It implies that, in the case of connected and low-degree $G_0$, the cardinality of $\mathbb{H}_{G_0,R}$ is close to the combination of all possible $\Omega_{G_0}(\mathbf{r})$ and $\mathbf{P}$ in Lemma 5. In other words, *the factorization given in Lemma 5 is nearly unique on $\mathbb{H}_{G_0,R}$.* From a pragmatic perspective, the result say that we can solve the constrained structure search issue from the factorization space as a alternative. More importantly, such the factorization space is independent to topology, because $G_0$ only determine the mapping $\Omega_{G_0}$, which is bijective, linear and fixed beforehand. The result guides us to find the practical solution on the graph-constrained structure search issue from the factorization space.

# 4 Encoding graph-constrained TN structures via a random-key trick

Inspired by the theoretical results, we introduce a practical coding method to embed the irregular TN structures into a regular discrete space, in which the population-based metaheuristics like GAs can be directly used for structure search. Last, experiments on a variety of benchmarks are implemented to demonstrate the effectiveness of the method.

## 4.1 Method

Figure. 2 depicts the coding process. We encode the elements of $\mathbb{H}_{G_0,R}$ from two ingredients: the rank-induced matrix $\Omega_{G_0}(\mathbf{r})$ and the permutation $\mathbf{P}$ as Lemma 5. For the former, since the mapping $\Omega_{G_0}$ is bijective and linear, the rank vector $\mathbf{r}$ of dimension $|E_0|$ is directly used as the code for this ingredient.

For the latter, we randomly embed $\mathbf{P}$ into the space $[0, 1]^{|V|}$, a set of decimal number vectors, by a *random-key* trick [4], which is popularly used to solve the optimal sequencing tasks. For the details, the random-key representation encode a permutation with a vector of random numbers from $[0, 1]$, and the order of these random numbers reflects the permutation. For instance, the code $(0.46, 0.91, 0.33)$ would represent the permutation $2 \to 3 \to 1$, by which we naturally have its matrix form $\mathbf{P}$. Finally, the encoded strings are simply the concatenation of the two ingredients.

One advantage of the random-key trick is robustness to the structure of $\mathbb{H}_{G_0,R}$. Regardless of the irregularity of $\mathbb{H}_{G_0,R}$, we always have the regular key space $[0, 1]^{|V|}$, on which the operations such as addition and perturbation are always available. It implies that the proposed coding method is $G_0$-independent, and many population-based metheuristics such as the one in [25] can be directly applied to graph-constrained structure search (see the numerical results given below.) .

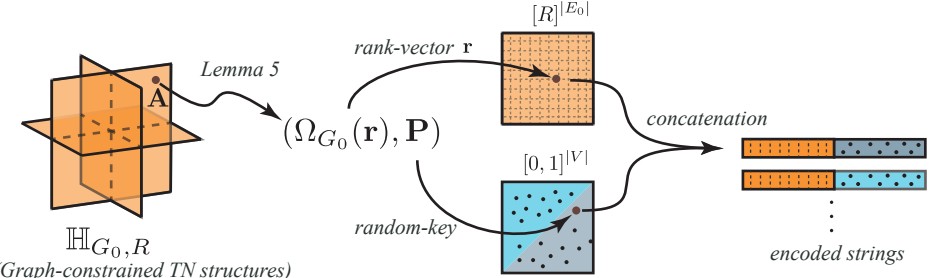

Figure 2: Illustration of encoding the graph-constrained TN structures into fixed-length strings. As Lemma 5, the structures are factorized by the rank-induced matrix $\Omega_{G_0}(\mathbf{r})$ and permutation matrix $\mathbf{P}$. In the method, $\Omega_{G_0}(\mathbf{r})$ is encoded by its non-zero entries, *i.e.* the rank-vector $\mathbf{r}$, into the space $[R]^{|E_0|}$ (the orange square). By the random-key trick, $\mathbf{P}$ is represented a vector of random number in the "key space" typically $[0,1]^{|V|}$ (the square with a mixed color in the figure). The final string is obtained by the concatenation of the two aspects. Note that, in the key space, different elements in the area with the same color represent the same permutation.

The proposed method gives more compact codes than the work in [25]. In the graph-constraint scenario, directly encoding the entries of the adjacency matrix as [25] cannot consider the "low-dimensional enssence" of $\mathbb{H}_{G_0,R}$ due to the irregularity. However, by the proposed method, the code length is shorted as $\mathcal{O}(|V|)$ compared to $\mathcal{O}(|V|^2)$ in [25]. A shorter code length implies faster convergence and lower computational requirement for the population-based methods in general. For the proposed method, we also prove the coding efficiency given in the supplementary material, which reflects the gap of the code length from the Shannon entropy on $\mathbb{H}_{G_0,R}$.

## 4.2 Numerical results

In this section, we evaluate the practical effectiveness and efficiency of the proposed coding method on various benchmark tasks for tensor network representation (TNR).

### 4.2.1 Searching the optimal TN structures on synthetic data in TR format and beyond.

In this experiment, we examine whether using the proposed coding method can learn sufficiently good low-dimensional representation on synthetic tensors in TR (including TT) format.

**Data generation.** We generate batches of tensors with randomly selecting TR structures. Specifically, we first let the dimension of each tensor mode equal 3. Then, we randomly generate the TR-ranks at discrete uniform distribution on $\{1, 2, 3, 4\}$ and the cores at Gaussian distribution $N(0, 1)$, and randomly permute the tensor modes after contracting the cores.

**Experiment setup.** The proposed coding method are directly applied to the genetic algorithm (GA) in [25] by replacing its chromosome design aspect, where we let $G_0$ be a cycle graph and the rank bound $R$ be equal to 7. Details of hyper-parameters on the GA are introduced in the supplementary material. For comparison, we also implement various types of TR decomposition methods with adaptive rank selection, which include the singular value decomposition (SVD) based method TR-SVD [47], least-squares-based method TR-ALSAR [47], Bayesian model Bayes-TR [35], and two general heuristics TR-LM [28] (exhaustive search) and TNGA [25] (population-based).

The experimental results are reported in Table 1, where the tensor order covers $\{4, 6, 8\}$ and the 5 generated tensors for each order are denoted as **Trial** A~E. For performance evaluation, we use the *Eff.* index [25], the ratio of number of parameters between the learned structures and the ground-truth TRs, to illustrate the model efficiency. We also illustrate the relative square error (RSE) and the generation (Gen.) of the optimal individuals in TNGA and ours in the table.

**Results.** As shown in Table 1, only our method can always achieve the same or lower-dimensional representation than the ground-truth. We observe that most of the TR decomposition methods *fail* dealing with the permutation on tensor-modes, and such the fact would limit the application of the TR methods in the practical use on high-order problems. We also observe the performance of

Table 1: Experimental results of searching structures on synthetic data in TR format. In the table, *Eff.* denotes the parameter ratio between the structures by different methods and the ground-truths; *RSE* in round brackets indicates the relative square error (ignored if smaller than $10^{-4}$.) and *Gen.* in angle brackets indicates the generation of the reported individual in TNGA and our method.

| Trial | Order 4 – *Eff.*↑ (*RSE*↓) ⟨*Gen.*↓⟩ | | | | | |
|---|---|---|---|---|---|---|
| | **TR-SVD** [47] | **TR-LM** [28] | **TR-ALSAR** [47] | **Bayes-TR** [35] | **TNGA** [25] | **Ours** |
| **A** | 1.00 | 1.00 | 0.21 | 1.00 | 1.00 ⟨004⟩ | 1.00 ⟨**003**⟩ |
| **B** | 0.64 | 1.00 | 1.00 | 0.64 | 1.00 ⟨002⟩ | 1.00 ⟨003⟩ |
| **C** | 1.17 | 1.17 | 0.23 | 1.00 | 1.17 ⟨005⟩ | 1.17 ⟨**003**⟩ |
| **D** | 0.57 | 0.57 | 0.32 | 1.25 (0.10) | 1.00 ⟨003⟩ | 1.00 ⟨**002**⟩ |
| **E** | 0.43 | 0.48 | 0.40 | 0.40 | 1.00 ⟨007⟩ | 1.00 ⟨**003**⟩ |

| Trial | Order 6 – *Eff.*↑ (*RSE*↓) ⟨*Gen.*↓⟩ | | | | | |
|---|---|---|---|---|---|---|
| | **TR-SVD** [47] | **TR-LM** [28] | **TR-ALSAR** [47] | **Bayes-TR** [35] | **TNGA** [25] | **Ours** |
| **A** | 0.21 | 0.44 | 0.14 (2e-3) | 0.25 (2e-3) | 0.82 ⟨011⟩ | **1.00** ⟨**010**⟩ |
| **B** | 0.14 | 0.15 | 0.14 | 0.44 (0.40) | 0.90 (6e-3) ⟨015⟩ | **1.00** ⟨**009**⟩ |
| **C** | 0.57 | 1.00 | 0.85 | 0.29 | 1.00 ⟨022⟩ | 1.00 ⟨**012**⟩ |
| **D** | 0.21 | 0.39 | 0.10 | 0.13 | 1.03 ⟨018⟩ | **1.16** ⟨**010**⟩ |
| **E** | 0.15 | 0.30 | 0.01 (0.02) | 0.12 | 1.00 ⟨016⟩ | 1.00 ⟨**007**⟩ |

| Trial | Order 8 – *Eff.*↑ (*RSE*↓) ⟨*Gen.*↓⟩ | | | | | |
|---|---|---|---|---|---|---|
| | **TR-SVD** [47] | **TR-LM** [28] | **TR-ALSAR** [47] | **Bayes-TR** [35] | **TNGA** [25] | **Ours** |
| **A** | 0.10 | 0.16 | 0.03 (0.20) | 0.03 | 0.48 ⟨017⟩ | **1.00** ⟨019⟩ |
| **B** | 0.09 | 0.43 | 0.06 (0.02) | 0.06 (7e-4) | 0.29 (2e-3) ⟨020⟩ | **1.02** ⟨**015**⟩ |
| **C** | 0.03 | 0.31 | 0.02 (0.01) | 0.02 | 0.49 ⟨015⟩ | **1.11** ⟨025⟩ |
| **D** | 0.20 | 0.53 | 0.02 (0.07) | 0.02 (0.02) | 0.32 ⟨027⟩ | **1.06** ⟨**013**⟩ |
| **E** | 0.33 | 0.33 | 0.02 (0.02) | 0.02 (3e-3) | 0.23 ⟨023⟩ | **0.88** ⟨**010**⟩ |

Table 2: Experimental results of searching structures on synthetic data in various TN format. In the table, *Eff.* denotes the parameter ratio between the structures by different methods and the ground-truths; *RSE* in round brackets indicates the relative square error (ignored if smaller than $10^{-4}$.) and *Gen.* in angle brackets indicates the generation of the reported individual of our methods. For rows, "ranks" means we fix the permutation part yet only learning the ranks, while "ranks+matching" means both the optimal ranks and permutation are learned.

| TNs | Our method | Trial – *Eff.*↑ (*RSE*↓) ⟨*Gen.*↓⟩ | | | |
|---|---|---|---|---|---|
| | | **A** | **B** | **C** | **D** |
| **T-Tree** [42] | *ranks* | 0.40 ⟨005⟩ | 0.41 (0.02) ⟨008⟩ | 0.40 (9e-3) ⟨006⟩ | 0.65 (0.04) ⟨005⟩ |
| | *ranks+matching* | 1.29 ⟨016⟩ | 1.17 ⟨014⟩ | 1.11 ⟨012⟩ | 1.55 ⟨012⟩ |
| **PEPS** [38] | *ranks* | 0.41 ⟨010⟩ | 0.43 (0.02) ⟨024⟩ | 0.39 (6e-3) ⟨027⟩ | 0.71 ⟨005⟩ |
| | *ranks+matching* | 1.14 ⟨013⟩ | 1.00 ⟨016⟩ | 1.00 ⟨007⟩ | 1.21 ⟨009⟩ |
| **H-Tucker** [16] | *ranks* | 0.49 (0.01) ⟨014⟩ | 0.64 ⟨010⟩ | 1.09 ⟨012⟩ | 0.81 ⟨006⟩ |
| | *ranks+matching* | 1.42 ⟨008⟩ | 1.21 ⟨023⟩ | 1.18 ⟨007⟩ | 1.29 ⟨011⟩ |
| **MERA** [11, 33] | *ranks* | 0.72 (0.01) ⟨012⟩ | 0.95 ⟨011⟩ | 1.93 ⟨011⟩ | 0.65 (0.04) ⟨014⟩ |
| | *ranks+matching* | 0.95 ⟨024⟩ | 1.32 ⟨008⟩ | 2.30 ⟨024⟩ | 1.00 ⟨027⟩ |

TNGA appears dramatically deterioration when increasing the tensor order. As analyzed at the end of Section 4.1, TNGA suffers from the dimension explosion of the search space. In this case, TNGA has to search the solution from about $4.6 \times 10^{23}$ candidates, which is almost $8.0 \times 10^{16}$ larger than the one of ours.

**TN structure search not limit to TR.** The proposed coding method is also useful for many well-known TNs in machine learning and physic not limit to TR. Under a similar setup for TR, we apply the proposed method to the TNs including T-tree (order-7) [42], PEPS (order-6) [38], hieratical Tucker (H-Tucker, order-6) [16] and multi-scale entanglement renormalization ansatz (MERA,

order-8) [11, 33]. Details of the data generation phase are given in the supplementary material. Table 2 illustrates the *Eff., RSE* and *Gen.* values by our method, where the rows of "ranks" mean we only learn the optimal TN-ranks while the rows of "ranks+matching" mean both the ranks and permutation are learned by our method. As shown in Table 2, our method achieves the TN structures as good as or even better than the ground-truth for various TNs. In addition, we also observe that a correct permutation on modes would significantly improve the representational power of TNs.

### 4.2.2 Benchmarks on real-world data

We consider three benchmark TNR problem on real-world data, where two of them is to represent the data and the other one is to represent learning models. Details of the experiment setup and more results are given in the supplementary material.

1. **Image compression.** We use GA equipped with the proposed coding method (in TR format) to compress 14 natural images randomly chosen from BSD500 [1], where images are grayscaled, resized by $256 \times 256$, and tensorized into order-8 tensors by two different tensorization: a "Python-like" reshaping operation denoted by "Trivial" and visual data tensorization (VDT) [6, 24, 45], a image-resolution-based tensorization method. As the result, we show the compression ratio (CR, in log form) and RSE (in round brackets) by the methods TR-SVD, TR-LM and ours in Table 3, and visualize the summary statistics of the learned permutation by our method in Figure 3.

2. **Image completion.** The same method is also implemented on image completion, a task to predict missing pixels from the observation. In the experiment, 8 images from USC-SIPI [40] are chosen and tensorized by VDT of order-9. After that, the entries are randomly removed at uniform distribution under the missing rate $\{50\%, 70\%, 90\%\}$, respectively. We show the average of RSE of predicting the missing values in Table 4 compared with the TT/TR completion methods TT-SGD [45], TRLRF [44], TRALS [39].

3. **Reparameterization of tensorial Gaussian process (GP).** TNR is applied to parameterizing the variational mean of GPs. In the experiment, we reparameterize the TT variational mean given in [20] by our method to search better structures. In a regression task on datasets CCPP [37], MG [14] and Protein [12], we have the TT variational mean of the order-$\{4, 6, 9\}$, respectively. In the result, we evaluate the performance by the number of parameters and mean square error (MSE, in the round brackets) shown in Table 5.

Table 3: Average of log compression ratio and RSE (in round brackets) for image decomposition.

|  | TR-SVD [47] | TR-LM [28] | Ours |
|---|---|---|---|
| Trivial | 0.95(0.14) | 0.94(0.14) | **1.35**(0.14) |
| VDT | 1.11(0.15) | 1.07(0.14) | **1.30**(0.14) |

(a)  (b)

Figure 3: Visualization of statistics on the similarity to the original permutation.

Table 4: Average of RSE on image completion under various missing percentage.

|  | TTSGD [45] | TRLRF [44] | TRALS [39] | Ours |
|---|---|---|---|---|
| 50% | 0.16 | 0.12 | 0.13 | **0.11** |
| 70% | 0.17 | 0.13 | 0.13 | **0.12** |
| 90% | 0.18 | 0.20 | 0.18 | **0.16** |

Table 5: Number of parameters and MSE (in round brackets) of GP regression under three datasets.

|  | CCPP | MG | Protein |
|---|---|---|---|
| **TTGP** [20] | 2640 (0.06) | 3360 (0.33) | 2880 (0.74) |
| **Ours** | **2244** (0.06) | **3008** (0.33) | **2032** (0.74) |

**VDT is verified as a more effective way for tensorization.** The results in Table 3 show that, our method owns higher compression ratio under close RSE compared to other methods. More importantly, the results show a significant difference when learning structures from two tensorization. Figure 3 illustrates the statistics on the similarity between the original permutation and the learned ones by our method. We observe from Figure 3(a) that in VDT the learned permutation is *significantly* closer to the original one than that in the "Trivial". Additionally, Figure 3(b) shows the cumulative distribution function (CDF), where we can see that, in VDT the probability is larger than $0.8$ for

the similarity being smaller than or equal to 2 . It implies that with a large probability the learned structures in VDT own at most a pair of permutation difference compared to the original one. On the contrary, for "Trivial" the probability is almost zero in the same interval. Hence, it is verified from the empirical results by our method that VDT is more effective way for image tensorization than the trivially reshaping operations.

**Exploring TN structures obtains lower-dimensional representation from incomplete data.** As shown in Table 4, our method achieve a comparable performance on the image completion task. Especially when the missing ratio is high, our method is forced to explore better TN structures not limit to the ranks, such that the lower-dimensional representation would be applied and results in more accurate prediction. Similar claims were also discussed in recent works [7, 17].

**Tensor-reparameterization: a potential way to compress learning models.** TNs are known as an efficient framework to compress learnables variables by low-dimensional cores. In the experiment, we illustrate from a "proof-of-concept" level that the model would be further compressed by *re*-parameterizing the learned TN in model. As shown in Table 5, we always use fewer parameters than its "teacher" model TTGP [20] to achieve the same MSE on the three datasets. It implies that our method give more efficient TNR by search better structures. Unlike training the model with simultaneously searching TN structures, we empirically find that searching better structures from the well-trained model in TN format would achieve better compression ratio. We intuitively conjecture that, by structure search, it is likely to obtain more efficient representation for a tensor, which has been in low-rank TN format. In the training phase, on the other hand, the models are not significantly low-rank in general. Therefore, the tensor reparameterization often gives better performance in practice. A rigorous analysis on this issue is still an open problem.

# 5    Discussion

Our experiments show good TN structures including ranks and permutations can be effectively learned in practice by the proposed coding method under extensive family of graph constraints, and our theoretical results show the the superior performance is thanks to the low-dimensional essence hidden behind the irregularity of the graph-constrained TN structures. More surprisingly, Proposition 9 shows that such the low-dimensional essence of TN structures is ubiquitous for most of practical TNs. As a consequence, we expect this work can promote the understanding on the structure search issue on tensor networks from both the theoretical and practical aspects, and the empirical claims in experiments are also expected to inspire more potential applications of TNs in machine learning.

**Limitation.** Theoretically, we only study the TNs, which do not contain the internal cores. Some well-known models like (H-)Tucker and MERA are not contained in the theory, although the proposed coding method works well for those models in experiments. Empirically, the proposed coding method is more suitable for the population-based methods like GAs, which are still computationally expensive compared to other heuristics. Also, the experiments on real-world benchmarks are only illustrative and proof-of-concept. More numerical results are necessary if stronger statements such as the performance improvement are expected.

# 6    Related works

Learning the optimal TN structures is a generalization of the rank selection issue for TN models [8, 9, 18, 26–28, 34, 43, 46, 47], and it is known as a tough task especially for the models that contain cycles in the topology [3, 23, 42]. More recently, there are several studies on learning TN structures [17, 19, 21, 25] in a more general form. Another line of works that are close to ours are studies focusing on the partition issue for H-Tucker decomposition [2, 13, 15], where the modes would be clustered to determine the optimal tree structure. Unlike them, this work is the first to solve the optimal matching problem as illustrated in Figure. 1. Moreover, we are the only few to theoretically study the structure search issue for tensor networks. From the algorithmic aspect, the random-key trick in our coding method is first proposed by [4], and popularly applied to solving difficult sequencing tasks such as the "travelling salesman problem" and the "clique problem" [31] in computational graph theory. Our method is also close to the subgraph search issue in the recent work [41], yet we focus on the different tasks and issues.

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
