# OpenReview forum: "Graph-Constrained Structure Search for Tensor Network Representation"
_NeurIPS.cc/2021/Conference — NeurIPS 2021 Submitted_

### Official Review · Reviewer_hLLS · 2021-07-15

**Rating:** 7
**Confidence:** 4

**Summary:**

The paper develop a means of identifying graphical structures which allow a tensor network (TN) to best fit a given input tensor. In contrast to previous work on this topic, a defining graph for the TN is also taken as input, and the optimal vertex permutation (i.e. an assignment of the vertices of the graph to the modes of the tensor) is learned, alongside an optimal choice of TN ranks for the graph in question. The paper gives both theoretical results characterizing the search space of valid TN configurations for this problem, as well as a new search method building on a previous genetic search algorithm [Chao Li and Zhun Sun 2020, "Evolutionary topology search for tensor network decomposition"] in this more limited setting. Experiments on synthetic tensors and real-world data show the efficacy of this new method in practice.

**Limitations And Societal Impact:**

The authors adequately address the limitations of their work in Section 5, and don't address potential negative societal impacts, owing to the technical nature of their work.

**Main Review:**

General architecture search for tensor networks has been the topic of several recent works (cited in the paper), and the present work singles out a special case of this problem which appears to be previously unexplored. Namely, the case where we have a given TN architecture of interest (e.g. tensor train, tensor tree, Tucker, etc.), and would like to find an optimal means of matching this architecture to an unknown data-dependent tensor. This question occurs frequently in applications of TNs to real-world problems, and previous approaches typically employ either physically-motivated heuristics, or else more general architecture search over arbitrary graphs. One of the main contributions of this paper is therefore the identification of this new intermediate regime, where graphical structures can be specified, and all other TN hyperparameters determined by algorithmic search. I suspect this special case will be of interest to TN practitioners.

The results provided by the authors are somewhat eclectic and hard to follow in places, and I personally found it difficult to understand the connection between the main theoretical results (Proposition 6 and Lemma 7, along with Propositions 8 and 9) and the proposed search method in Section 4. While some justification of the connection between these two families of results is given in the manuscript (and I am satisfied that they are indeed related), it would be helpful to readers to spend more time motivating the new search method, which is currently introduced rather abruptly. Crucially, it would be very helpful to give a more detailed introduction to the random key trick employed, which is currently described in a single paragraph, with much of the explanatory work deferred to a reference on genetic algorithms. More generally, I would also encourage the authors to give a quick overview of the genetic search algorithm on which the current architecture search method is built [Chao Li and Zhun Sun, 2020], since it is relevant for understand the experiments.

**Time Spent Reviewing:**

3

---

> ### Author Response · Authors · 2021-08-09
> **Response to Reviewer hLLS**
>
> We appreciate the insightful comments by the reviewer. The reviewer acknowledges that the proposed work is novel and of interest to tensor network (TN) practitioners, and the main concern is insufficient discussion on methodology and the connection between the theoretical and practical aspects of the work.
>
> **The connection between theoretical results and the search method.** Thanks for the suggestion. In the revised manuscript, we will emphasize this connection to enhance the motivation of the new search method. Below, please allow us to highlight the purpose of introducing each theoretical result ("Proposition 6 and Lemma 7, along with Propositions 8 and 9" as mentioned) and the connection with the search method:
>
> - Proposition 6 is to *demonstrate and understand* why the exiting search methods like GA (Li and Sun, 2020) or greedy method (Hashemizadeh et al., 2020) cannot tackle the constrained search issue at the current stage.
> - Proposition 8 (and Lemma 7) is to quantitatively analyze how the symmetries (a topology-related property) impact the cardinality (scale) of  the search space. The result reveals that the impact by the symmetries is not as strong as we intuitively expected for many practical TNs, and therefore motivates the study on a universal cardinality bound (Proposition 9).
> - Proposition 9 formally proves this point in a more general form. More importantly, this claim allows us to design the search method in a very simple and efficient manner, i.e., it is sufficient to encode the TN structures by Eq. (5) and the symmetries are proved to be ignorable without much loss of efficiency in practice.
>
> We would also like to highlight the novelty of the work (please see the paragraph "**Novelty of the paper**" at the response to *Reviewer U3Vg*). More discussion on the methodology like "random-key" will be also carefully addressed in the revised version according to the suggestion.

---

### Official Review · Reviewer_4rtP · 2021-07-16

**Rating:** 8
**Confidence:** 4

**Summary:**

	This paper shows that the graph constrained TNs are irregular, which makes it tricky for the conventional search algorithms on grids to obtain feasible solutions. Under this situation, according to universal cardinality bound across a series of practical TNs, this paper construct regular discrete space, representing those irregular TNs in a compact manner. At last, this paper proposes the “random-key” trick to encode TN structures into fix-length strings(i.e., coding space). In the regular coding space, population-based algorithms with the proposed coding method can search TNs effectively. And the experiments demonstrate the good performance of the proposed methods.

**Main Review:**

To solve the searching problems on these structures, this paper has provided the following contributions.

1. Formulating this graph constrained TNs is reasonable since that this formulation incorporating plenty of tensor formats and a wide searching space implies a higher probability to find a suitable structure.
2. The analysis on the universal bound is interesting, which instructs the process of transferring an irregular space to a regular space.
3. This paper presents a coding method named as random-key to transform the irregular space. And experiments show that it works well.
4. This paper is technically sound.

However, there still exist some issues to be addressed.

1. In Line 59, $\mathcal{X}_{i_1, i_2, \dots, i_N}$  denotes an element rather than a tensor.
2. In Definition 1, for easy following, it is better to change the suffix $R$ of  $\mathbb{A}$  to ${R+1}$
 for consistent with $\mathbb{Z}_{R+1}$.
3. Experiments seem mostly conducted on common tensor formats like the tensor ring and the tensor train. It will be better to dicuss some special format like the odd tensor in [1].
4. Although I can find many more experiments in the supplementary material, it suggests adding some more SOTA methods in image completion for comparison for  completeness.
5. To be more clear, it is helpful to add more details of random-key.
6. There are some writing typos:
   (a) In Line 155, "it" -> "It";
   (b) In Line 190, "say" -> "says";
   (b) etc..

Overall, for the novel idea and interesting method, I prefer to give acceptance to this paper.

[1] Li, Chao, and Zhun Sun. "Evolutionary topology search for tensor network decomposition." International Conference on Machine Learning. PMLR, 2020.

**Time Spent Reviewing:**

5

---

> ### Author Response · Authors · 2021-08-09
> **Response to Reviewer 4rtP**
>
> We appreciate the comments given by the reviewer. In the comments, the reviewer agrees that *(i)* the formulated problem is reasonable and the idea is novel; *(ii)* the universal bound (Proposition 9) is interesting and *(iii)* the experimental results are good. The reviewer also addressed  several revision suggestions, which mainly focus on notation and additional experimental results.
>
> **Additional experiments on special format of tensor networks.** Thanks for the suggestion. It will be addressed in the revised version of our work. In the existing experiments, we have evaluated the effectiveness of the proposed method  covering the models of TT (TR), T-tree, PEPS, H-Tucker and MERA. We therefore expect that it would work well in more specific models like the one in (Li and Sun, 2020).
>
> **The image completion task in the paper.** To fairly evaluate if the structure search has capability of improving the completion performance, we fixed the graph constraint as tensor ring (TR) and  therefore purposely selected the baselines, which also apply TR as the backbone model. Richer results compared with more sophisticated methods will be given in a complete version of this work.
>
> We also thank the helpful and detailed suggestions on the readability of the manuscript. They will be carefully revised.

---

### Official Review · Reviewer_U3Vg · 2021-07-16

**Rating:** 4
**Confidence:** 3

**Summary:**

This paper considers the problem of selecting ranks and permutations of a given tensor network structure. It analyzes the size and some properties of the resulting search space, proposes a simple representation, plugs it into a genetic algorithm, and reports some experimental results.


**Limitations And Societal Impact:**

Ok

**Main Review:**

Generally, the ideas put forward in this paper are very simple (which is good), but they are presented poorly (1,2). The novelty and amount of insight of this paper is low (2,3,4); it basically boils down to a small modification of [25]. Nevertheless, the experimental study shows promising results, but it's too "proof-of-concept" to be really insightful (5, acknowledged by the authors themselves). For these reasons, I do not recommend to accept the paper, even though it shows a promising direction.

1. The presentation of this paper lacks precision and clarity from time to time. For example, Def 1 is imprecise and consequently confusing,  Prop 6 refers to "a probability being approximately larger", and Prop 9 to a "constant far less than |V|". Clear, self-contained definitions would make this paper more accessible. That being said, the main points become clear even though I am not an expert in this particular area.

2. The key idea of this paper is to start with a template of a tensor network and then search the space of possible instantiations (rank to each edge + permutation of vertexes). When phrased like this, many of the points raised in Sec. 3 are straighforward, if not trivial (e.g., Lemma 5+7) or unnecessarily (Prop 6). The authors complicate things here by arguing around the adjacency matrix of the template instead of directly the template graph. Since also the representation of a tensor network constraint in terms of a graph is not novel, I did not find this discussion particularly insightful or novel.

3. Similar remarks apply to the encoding scheme of section 4. It's straightforward; the key idea is to enumerate the edge labels in a fixed order and to represent the permutation via order statistics of real numbers associated to each vertex ("random-key trick"). The method of [25] is then employed. Overall, this appears to be a small (but admittedly beneficial) improvement over [25].

4. What I find surprising, if not disappointing, is that the authors argue about the need to account for symmetries in the first part of the paper, but then ignore symmetries completely in the second part and their ultimate method. Here they are "hiding" behind the O-notation of Prop 9, which (i) may have large constants that are very relevant in practice and (ii) only apply to templates in which the degree is bounded by a constant.

5. The experimental study suggests that the methods produce better results than prior works in term of compression ratio. The experimental setup, however, is not convincing. The authors list a large number of hyperparameter choices in the appendix, but do not state how these hyperparameters have been been selected. The choice of synthetic datasets is not well argued for and they appear small (i.e., dim 3 per mode).

UPDATE AFTER AUTHOR FEEDBACK

I've read the response of the authors. My overall impression remains the unaffected. As mentioned in the original review, I do think that this paper has potential. All of the concerns raised in my original review remain, however.

Some comments on the response:

The analysis is not very deep and can be presented in simpler terms using the TN template. In fact, the presentation in terms of an adjacency matrix does not add anything other than making the discussion more technical. Note that that's different for [25] because there the graph structure itself is encoded via its adjacency matrix during TN search. But in this submission, the structure is fixed up to permutations and ranks.

Prop. 9, which the authors cite in their response, is not clear as stated. It's of the form "x>=O(...)", which is confusing at best: ">=" denotes a lower bound, but O(...) an upper bound. Be that as it may, it would be more helpful if the authors provide bounds that explicitly involve the maximum degree.

Exaggerating somewhat, the authors seem to argue that (i) symmetries are very important so that prior methods don't work and (ii) symmetries can be ignored in their method. I do understand what the authors actually want to say, but it's not clearly argued in the write-up.

Similar remarks as for the analysis apply to the actual algorithm. If expressed in terms of graphs, it boils down to search on rank per edge and position per vertex. The use of the random-key trick for the latter is well-known, as the authors acknowledge. There is nothing wrong with this method, but there is also not much novelty and it does ignore all symmetries.

It's not clear what the authors mean by "applied the same genetic operators as [25]". Clearly, [25] has different operators; e.g., it changes the graph structure.

Decent hyperparameter search and a clear description of the search process across all methods is key for reproducibility and fairness. Likewise, the experimental setup needs to be argued for clearly.


**Time Spent Reviewing:**

3

---

> ### Author Response · Authors · 2021-08-09
> **Response to Reviewer U3Vg**
>
> Thanks for the comments by the reviewer. In the comments, the reviewer acknowledges that the work shows *a promising direction*, *the idea is very simple* (which is good) and there is *admittedly beneficial improvement* on the performance. The main concern of the reviewer mainly focuses on *the presentation and novelty* of the paper.
>
> **Novelty of the paper.** The reviewer mainly doubts the novelty of our work from three aspects: *(i)* the theoretical results are straightforward (Q2), *(ii)* the proposed encoding scheme is straightforward as well (Q3); and *(iii)* shallow insight of Proposition 9 (Q4).  Here we would like to highlight our main contribution from those mentioned aspects: theoretically, we are ***the first to rigorously analyze the symmetry property of the tensor network (TN) structures*** (Lemma 7, Proposition 8); practically, the proposed scheme is ***the first and unique** so far* to tackle the permutation issue of TN structure search, and ***decreases the size of the search space significantly*** from $\mathcal{}$$\mathcal{O}(\vert{}V\vert^2)$ [25] to $\mathcal{O}(\vert{}V\vert)$ for most of TN models; and Proposition 9 (a universal cardinality bound on the search space) reveals a non-trivial fact that ***the cardinality (size) of the search space is almost topology-independent*** for most of practical TNs, such as TT, TR and PEPS, due to the low degree nature of those models. It allows us to design the practical method in a very simple and efficient manner since ***the symmetries are proved to be ignorable under practically reasonable conditions*** (mentioned by the reviewer in Q4 as it is surprising).
>
> **The presentation of the paper.** Thanks for the helpful suggestions and we will carefully improve the presentation of our work. In the revision, we will improve the self-containedness of the manuscript for a wider range of readers and illustrate the rigorous and complete form of claims not only in the proof. Below we response each comment, respectively.
>
> *1.  "The presentation of this paper lacks precision and clarity from time to time. For example, Def 1 is imprecise and consequently confusing, Prop 6 refers to "a probability being approximately larger", and Prop 9 to a "constant far less than |V|"*
>
> — Thanks for pointing out. They will be carefully improved. In your mentioned cases, the explicit probability bound in Proposition 6 is given by Eq. (5) in line 53 of the supplementary material (Supp); and in Proposition 9 the condition "constant far less than $\vert{}V\vert$" stands for "$\Delta_{G_0}\ll{}\vert{}V\vert$", which is sufficient to prove the inequality (7).
>
> *2-1. "The key idea of this paper is to start with a template of a tensor network and then search the space of possible instantiations (rank to each edge + permutation of vertexes). When phrased like this, many of the points raised in Sec. 3 are straighforward, if not trivial (e.g., Lemma 5+7) or unnecessarily (Prop 6)."*
>
> — We highlight the aim and contribution of addressing those results in the paper as follows. Lemma 5 is *necessary* to derive the proposed encoding method in Section 4.  Lemma 7, to the best of our knowledge, is the *first* to quantitatively reveal the symmetric property when searching the TN structure and a required lemma to have Proposition 8.  Proposition 6 is used to *demonstrate and understand* why the existing SOTAs like GA [25] or greedy methods like [17] cannot tackle the targeted issue.
>
> *2-2. " The authors complicate things here by arguing around the adjacency matrix of the template instead of directly the template graph. .Since also the representation of a tensor network constraint in terms of a graph is not novel, I did not find this discussion particularly insightful or novel"*
>
> — The adjacency matrix is an elegant way to model the tensor network structures, which uniformly include the both topology and rank information. Not only has the same idea been applied in [25], but the adjacency matrix is more natural form than graphs to derive the encoding method proposed in Section 4. Meanwhile, such a matrix formulation on TN structures does not complicate the main theoretical results (such as Proposition 6, 8 and 9) in the work. Although the graphical representation of TNs has been known in existing literature, *we are the first to analyze the TN structure search issue by tools from the algebraic graph theory*.
>
> *3. "Similar remarks apply to the encoding scheme of section 4. It's straightforward; the key idea is to enumerate the edge labels in a fixed order and to represent the permutation via order statistics of real numbers associated to each vertex ("random-key trick"). The method of [25] is then employed. Overall, this appears to be a small (but admittedly beneficial) improvement over [25]."*
>
> — The proposed structure encoding scheme is good (acknowledged by the reviewer) and efficient to solve the targeted search problem. As shown in Table 1 (especially in the high-order case), the improvement by the proposed encoding is significant compared to the SOTA [25]. More importantly, only the search method equipped with the proposed encoding has the capability of tackling the "permutation" task, and such the characteristic is uniquely to this date. From a methodological perspective, in the work we focused on developing more efficient TN structure encoding method, and therefore applied the same genetic operators as [25] for reasonable and fair evaluation.
>
> *4-1. "What I find surprising, if not disappointing, is that the authors argue about the need to account for symmetries in the first part of the paper, but then ignore symmetries completely in the second part and their ultimate method. "*
>
> — Proposition 9 reveals that the symmetries are ignorable in practice without much loss of the efficiency yet achieving a simple and universal encoding scheme. We have discussed this point at the paragraph "**Novelty of the paper**", and several instances such as TT, TR and PEPS given in *Proposition 8* can intuitively support this claim. Computationally, to test a non-trivial automorphism for a general graph is also known as a difficult task (NP-intermediate) . We therefore ignore the symmetries in the proposed encoding scheme due to the limited benefit and possible expensive computation.
>
> 4-2. "Here they are "hiding" behind the O-notation of Prop 9, which (i) may have large constants that are very relevant in practice and (ii) only apply to templates in which the degree is bounded by a constant."
>
> — (i) As shown in (6) in Supp, we do *not* find large constants that are very relevant in practice. (ii) Yes. The condition "its ($G_0$) maximum degree $\Delta_{G_0}$ is a constant that is far less than $\vert{}V\vert$" is *necessary* for the establishment of Proportion 9, and is reasonable in practice (see the discussion in line 182-185 of the manuscript and Table 1 in Supp).
>
> *5-1. "The authors list a large number of hyperparameter choices in the appendix, but do not state how these hyperparameters have been been selected."*
>
> — We select the hyperparameters according to the suggestions by [25]. To be fair, we use the same hyperparameter configuartion for the both TNGA [25] and ours in the experiment. We applied the exhaustive search to  practically find the optimal hyperparameters for other baselines.
>
> *5-2. "The choice of synthetic datasets is not well argued for and they appear small (i.e., dim 3 per mode)."*
>
> — For fair evaluation, we constructed the synthetic data according to the suggestions by [25], where they chose the dimension per mode equaling 2 (while we chose dim 3 per mode in our experiment). As the tensor order has more impact than the dimension per mode to determine the difficulty of the task. We therefore focused on evaluating the methods under different tensor orders yet fixing the dimension per mode.

---

### Official Review · Reviewer_mrqp · 2021-07-28

**Rating:** 5
**Confidence:** 3

**Summary:**

This paper addresses a new problem when searching for an optimal structure in the tensor network representation on the condition that the graph's topological structure is constrained and isomorphic to a specific graph G_0.
TNR is more flexible than the classical tensor decomposition in terms of its high freedom of how the multiplication among multiple core tensors is done. However, if a specific graph structure is preferred such as TT or TR, when representing the topological structure of TNR by an adjacency matrix, the adjacency matrix may not be the best representation form due to its inability to identify those isomorphic graphs. The proposed work firstly attempted to prove that there exists a bijective mapping from a rank vector r to any graph in an automorphic set. Then, the permutation vector p from the bijective mapping concatenated with r is an encoding vector of a specific graph in the automorphic set. The encoding vector can be used in the later search problem via GA or other population-based algorithms.
Some experiments were done to prove the efficacy of the encoding by showing that via the proposed encoding method GA can find the solution quicker and better compared with other baseline methods.



**Limitations And Societal Impact:**

I identified no potential negative societal impact of their work. The limitations are well discussed in Section 5.

**Main Review:**

The topic is of significance and interest. A new topological constraint problem is formulated. However, the clarity of the paper is one concern of mine. The paper could be much improved its clarity and make it more self-contained.
Detailed comments:
1. ln 64, C_i,k = A_i,j B_j,k -> Should there be a sum on the RHS?
2. ln 76, "with the ranks upper-bounded by R" -> The rank is not well-defined in the Def 1. It confuses me whether the rank is the deg(v) or the rank(unfold(X)).
3. ln 77, internal cores are not defined.
4. ln 83, "corresponds" -> "correspond".
5. ln 84, "...will show the matrix form..." -> "...will show that the matrix form".
6. Please try to make the definitions clearer as [25] did.
7. ln 92, "...with N vertices and a edge..." -> "...an edge..."
8. ln 132, Omega_G_0 and r are defined but not constructed explicitly. Please try to clarify how to obtain r from the graph G_0 and the detailed definition of r, and please explain what the rank vector of dimension |E_0| means. How is P computed?
9. In the Supplemental Materials, what is the definition of H_0^u in the proof, and what is the definition of Phi^{-1} and Phi? And where is the Omega_G_0 in the proof of Lemma 5? Please make sure that the proof is completed and correct.
10. ln 141, what does that mean by "relatively sparse"? please be specific. And what does it mean by "not closed with a probability"? Should it be a deterministic property when we talk about a set's closeness?
11. ln 157, "TNs with symmetric topologies like TR and the complete TN are expected to own a smaller size of H_G0,R" -> I feel it is not convincing to me since CTN may increase |E_0| in Eq (6) shown in Lemma 7, which could significantly increase the size of H.
12. ln 190, what does it mean by "nearly unique"?
13. ln 240, "the tensor order covers {4,6,8}" -> Do the authors mean tensors of order 4, 6, 8 are generated?
14. In many real-world cases, the underlying topology is more complicated than TR or TT, and limited prior knowledge is known, so I am curious if the proposed method can generalize well?
15. What is the coding format of the image completion experiments?



**Time Spent Reviewing:**

72

---

> ### Author Response · Authors · 2021-08-09
> **Response to Reviewer mrqp**
>
> Thanks for the detailed and helpful comments given by the reviewer. The reviewer agrees with the significance and novelty of the formulated problem and the main concern is the clarity of the paper.
>
> **Importance of the topic.** We appreciate the acknowledgement of the significance of our formulated new problem, i.e., the constrained structure search for tensor network (TN) representation. Indeed, ***the problem occurs frequently in applications of TNs*** to real-world applications (also acknowledged by Reviewer 4rtP and hLLS), especially in the machine learning community. To the best of our knowledge, the proposed method is ***the first and unique so far to tackle the issue*** (also acknowledged by Reviewer hLLS) . The experimental results also demonstrate the ***superior performance compared to the existing SOTA [25]*** and the potential capability in wide range of applications.
>
> **Clarity of the manuscript.**  We agree with this point and will carefully improve the clarity of our work. We summarise the comments into three categories:  incomplete review on the existing concepts (1-3) such as TN ranks, unclear presentation in theoretical claims and proofs (8-12)  and typos (4-7).  In the revision, we will add an additional paragraphs for notation and concepts, illustrate the complete form of claims not only in the proof, and carefully recheck the notable typos and grammar errors of the manuscript. Below, we response each question, respectively, and the indices below keep the same to those in the comments.
>
> 1. "*ln 64, C_i,k = A_i,j B_j,k -> Should there be a sum on the RHS?"*
>
> — Yes, there should be a summation over $j$ in this case. We omitted the sum symbol according to the Einstein summation convention, which is widely used in tensor network literature and python libraries.
>
> 2. "*ln 76, "with the ranks upper-bounded by R" -> The rank is not well-defined in the Def 1. It confuses me whether the rank is the deg(v) or the rank(unfold(X))."*
>
> — The "ranks" here stands for the tensor network (representation) ranks [30, 47], (or bond dimension in physics), which correspond to the collection of the labels for each edge in our graphical model.
>
> 3. "*ln 77, internal cores are not defined."*
>
> — The internal cores stands for a class of TN's core tensors, which have no "free legs" such as the central tensor in Tucker decomposition.
>
> 4-7."*ln 83, "corresponds" -> "correspond"."* "*ln 84, "...will show the matrix form..." -> "...will show that the matrix form"."* "*Please try to make the definitions clearer as [25] did."* "*ln 92, "...with N vertices and a edge..." -> "...an edge...""*
>
> *— Thanks for pointed out. The typos will be carefully revised.*
>
> 8-1. "*ln 132, Omega_G_0 and r are defined but not constructed explicitly. Please try to clarify how to obtain r from the graph G_0 and the detailed definition of r, and please explain what the rank vector of dimension |E_0| means. "*
>
> — The explicit form of $\Omega_{G_0}$is given in line 35 of the supplementary material (Supp), and the construction of the rank vector $\mathbf{r}$ is given in line 32-33 of the Supp.
>
> 8-2. *"How is P computed?"*
>
> — The explicit factorization $\mathbf{H}=\mathbf{P}\Omega_{G_0}(\mathbf{r})\mathbf{P}^\top$ is *not required* in the work. Lemma 5 is to prove that all $\mathbf{H}$ in the search space can be represented by Eq. (5), i.e.,  *to prove the feasibility of the factorization*. In the method, we encode different $\mathbf{H}$ directly by giving  $\mathbf{P}$ and $\mathbf{r}$ different values.
>
> 9. "*In the Supplemental Materials, what is the definition of H_0^u in the proof, and what is the definition of Phi^{-1} and Phi? And where is the Omega_G_0 in the proof of Lemma 5? Please make sure that the proof is completed and correct."*
>
> — In the proof, we define $\mathbf{H}^{u}$ by the equation  $\mathbf{H}^{u}=\mathbf{P}\mathbf{H}_0\mathbf{P}^\top$  *in line 26 of the Supp,* where $\mathbf{P}$ denotes the permutation matrix and $\mathbf{H}_0$ denotes the adjacency matrix of $G_0$.
>
> The Phi (we guess what you mean is the mapping $\Psi$ using in the proof ) is *defined in  Lemma 2* (given in line 91 of the manuscript) and $\Psi^{-1}$ stands for its inverse (since $\Psi$ is bijective).
>
> The $\Omega_{G_0}$ satisfies  $\Omega:\mathbf{z}\mapsto\Psi^{-1}(G_0,g^{-1}(\mathbf{z}))$ given in line 35 of the Supp, where the mapping $g$ is defined in the proof and the sub-script of $\Omega_{G_0}$ is omitted  in the proof without confusion for brevity.
>
> 10-1. "*ln 141, what does that mean by "relatively sparse"? please be specific. "*
>
> — Intuitively, the "relatively sparse" implies a small number of edges of a graph.
>
> The explicit form of the probability bound with a *$(k,l)$-sparse graph $G_0$* is given by Eq. (5) in the Supp.
>
> 10-2. "*And what does it mean by "not closed with a probability" (in* Proposition 6 (claim TWO) *)? Should it be a deterministic property when we talk about a set's closeness?*"
>
> — In Proposition 6 (claim TWO), we discussed if a random perturbation operation on the candidate set is closed, i.e., if a TN structure still satisfies the constraint after perturbation.
>
> The property is probabilistic due to the random nature of the perturbation.
>
> *It is different from studying the openess (or closeness) of a set*.
>
> 11. "*ln 157, "TNs with symmetric topologies like TR and the complete TN are expected to own a smaller size of H_G0,R" -> I feel it is not convincing to me since CTN may increase |E_0| in Eq (6) shown in Lemma 7, which could significantly increase the size of H."*
>
> — Sorry for our confusing words. You are right. The CTN model owns a huge size of the candidate set as you mentioned. It will be expressed more clearly.
>
> 12. "*ln 190, what does it mean by "nearly unique"?"*
>
> — It means that given a $\mathbf{H}$, there exist one and *almost* only one pair $(\mathbf{P,r})$ such that Eq. (5) is held.
>
> Strictly, it is not unique  due to the symmetry of the TN structures, i.e., different permutations would lead to the same structure.
>
> However, Proposition 9 implies that the ambiguity by symmetry can be practically ignored if $G_0$ is with low graph degree.
>
> We therefore claimed that the factorization Eq.(5) is nearly unique.
>
> 13. "*ln 240, "the tensor order covers {4,6,8}" -> Do the authors mean tensors of order 4, 6, 8 are generated?"*
>
> — Yes. In the experiment, we generated the synthetic tensor of order 4,6,8, respectively. The experimental results are given in Table 1.
>
> 14. "*In many real-world cases, the underlying topology is more complicated than TR or TT, and limited prior knowledge is known, so I am curious if the proposed method can generalize well?"*
>
> — We agree. We focused on the two models in the real-world case because *TT and TR are popularly used and there exist rich baseline methods for suitable numerical evaluation*.
>
> The proposed method has the capability of modeling many complicated tensor networks by pre-defining different topology $G_0$ such as T-tree, PEPS and MERA (cf. Table 2).
>
> We expect that the proposed method owns a good generalization from the learning perspective because it is always to represent a problem with fewer parameters (to avoid the potential over-fitting issue) than the conventional tensor network methods.
>
> 15. "*What is the coding format of the image completion experiments?"*
>
> — We guess what you concern is the structure encoding method in image completion. In this task, we choose the tensor ring (TR) as the backbone $G_0$ and learn the optimal ranks and permutations by GA equipped with the proposed encoding method.
>
> The encoding method is same to the one in image compression and etc..
>
> The difference of the model settings is that in image completion we use the tensor network to only fit the observable pixels, while in image compression all pixels would be fitted.

---

### Decision · Program_Chairs · 2021-09-27

**Decision:**

Reject

**Comment:**

The problem tackled by this paper, searching for graph constrained tensor network structures, is relevant and potentially impactful, and the core ideas presented in the paper are potentially useful for the intersection of the tensor network and ML communities. However, through the discussion, the reviewers overall agreed that the paper is not yet ready for publication and that a substantial revision of the submission is needed before being published to make sure that the paper obtains the potential impact it deserves. The most important points that were raised in the reviews and the discussion are the following:

- The theoretical analysis seems overall disconnected from the practical/algorithmic contribution of the paper. While it is true that, for an expert in TN methods for ML, the theoretical analysis makes sense and is relevant, these results are presented in a somehow convoluted way that is likely to be an obstacle for the interested readers to benefit from the paper. I recommend the authors to thoroughly go through the reviews and to try to clarify the relevance of the theoretical analysis for the contributions of the paper. I believe revising the paper taking into account the reviews will make the paper more impactful once it is published.

- The novelty of the paper needs to be better argued and demonstrated.

- Several reviewers mentioned that the mathematical exposition needs to be more rigorous in several places.

In summary, the paper is not yet ready for publication mainly due to lack of rigor in the mathematical exposition, a lack of clarity on the connection between the theoretical results and the proposed algorithm, and the fact that the method could be presented in a simpler manner (which however could potentially question the novelty of the proposed approach, wrt [25] in particular). That being said, the problem considered by the authors is interesting and, if reworked and better formalized, the theoretical results could be relevant as well.